# Molecular dynamics study on the effects of charged amino acid distribution under low pH condition to the unfolding of hen egg white lysozyme and formation of beta strands

Husnul Fuad Zein[1,2,3], Ibrar Alam[1], Piyapong Asanithi[1], Thana Sutthibutpong[1,2,3,4]*

1 Nanoscience and Nanotechnology Program, Faculty of Science, King Mongkut's University of Technology Thonburi (KMUTT), Thung Khru, Bangkok, Thailand, 2 Department of Physics, Theoretical and Computational Physics Group, KMUTT, Thung Khru, Bangkok, Thailand, 3 Faculty of Science, King Mongkut's University of Technology Thonburi (KMUTT), Thung Khru, Bangkok, Thailand, 4 Center of Excellence in Theoretical and Computational Science (TaCS-CoE), Faculty of Science, King Mongkut's University of Technology Thonburi (KMUTT), Thung Khru, Bangkok, Thailand

* thana.sut@mail.kmutt.ac.th

**Data Availability Statement:** All relevant data are within the paper and its Supporting Information files.

## Abstract

Aggregation of unfolded or misfolded proteins into amyloid fibrils can cause various diseases in humans. However, the fibrils synthesized in vitro can be developed toward useful biomaterials under some physicochemical conditions. In this study, atomistic molecular dynamics simulations were performed to address the mechanism of beta-sheet formation of the unfolded hen egg-white lysozyme (HEWL) under a high temperature and low pH. Simulations of the protonated HEWL at pH 2 and the non-protonated HEWL at pH 7 were performed at the highly elevated temperature of 450 K to accelerate the unfolding, followed by the 333 K temperature to emulate some previous *in vitro* studies. The simulations showed that HEWL unfolded faster, and higher beta-strand contents were observed at pH 2. In addition, one of the simulation replicas at pH 2 showed that the beta-strand forming sequence was consistent with the 'K-peptide', proposed as the core region for amyloidosis in previous experimental studies. Beta-strand formation mechanisms at the earlier stage of amyloidosis were explained in terms of the radial distribution of the amino acids. The separation between groups of positively charged sidechains from the hydrophobic core corresponded to the clustering of the hydrophobic residues and beta-strand formation.

## 1. Introduction

Lysozymes are a family of globular enzymes in the immune systems of animals. A lysozyme molecule is a single polypeptide of around 130 residues that can partially hydrolyze the peptidoglycans of gram-positive bacterial cell walls [1, 2]. Lysozyme is one of the protein types associated with the formation of amyloid fibrils, as the failure of specific peptides or proteins to fold or to remain correctly folded triggered non-functional protein aggregation [3]. Amyloid

**Funding:** This research project was supported by the Thailand Science Research and Innovation (TSRI) Basic Research Fund through a grant awarded to TS. The Petchra Pra Jom Klao Doctoral Scholarship, KMUTT, also provided support to HFZ.

**Competing interests:** The authors have declared that no competing interests exist.

aggregation is a hallmark of several degenerative diseases, such as Alzheimer's disease, Parkinson's disease, type II diabetes, Creutzfeldt Jakob, Huntington's, amyotrophic lateral sclerosis (ALS) [4, 5]. However, non-toxic forms of amyloid fibrils could be useful in some areas of applications, such as bioengineering, biosensor, drug delivery, regenerative medicine, cell-encapsulating materials, tissue engineering, molecular and electronic devices, etc. [6–17]. Amyloid fibril formation *in vitro* can be controlled under several factors that either stimulate or inhibit aggregation [18]. Hen egg-white lysozyme (HEWL) is one the most commonly used proteins in protein aggregation research due to its well-characterized structure and low cost. A high temperature and an acidic condition can accelerate the unfolding and aggregation of HEWL, as a previous study showed that amyloid fibrils were formed when incubating HEWL under the temperature of 65°C and pH 2 for 196 hours [1].

Conformational changes of the unfolded and aggregated proteins can be monitored by using several methods, such as atomic force microscopy (AFM), Raman spectroscopy, and electrochemical impedance spectroscopy (EIS) [19–21]. Under an elevated temperature and acidic condition, AFM could provide images of lysozyme aggregation at various incubation times by depicting the time-dependent sizes of self-assembled spheroidal oligomers. Changes in both secondary and tertiary structures of aqueous lysozymes could also be observed by shifting Raman spectra in the range of 650–1875 cm$^{-1}$ from its native state [1]. Signals from EIS spectroscopy are highly sensitive to structural changes of proteins due to their unique charge transfer resistances (Rct) [21]. These methods, along with dynamic light scattering, size exclusion chromatography, transmission electron microscopy (TEM), surface plasmon resonance (SPR) [1, 22, 23], have their advantages but still lack understanding of the molecular details of protein structural changes.

Computer simulations have become an alternative tool to provide more insight into the structures and dynamics of proteins at different states under different conditions in atomistic details. The accuracy of atomistic molecular dynamics (MD) simulations to predict the molecular behavior of proteins under extreme conditions [24, 25] has been improved with the continuing development of molecular mechanics forcefield parameters [26, 27]. Therefore, some computational studies were conducted to observe the effects of temperature, solvents, and external perturbation on the unfolding and aggregation of human lysozymes and HEWL [25, 28–34]. From MD simulations performed by Moraitakis *et al.*, replacing an aspartic acid residue with a histidine caused human lysozyme to unfold significantly faster under high temperature and could reproduce an experimental result. The mutation was proposed as a possible seed for amyloidosis [25]. Jafari *et al.* demonstrated in molecular detail that the lysozyme unfolds better in high concentrations of the sodium dodecyl sulfate (SDS) surfactant at 370 K, higher than the thermal denaturation midpoint temperature (Tm) [28]. Jiang *et al.* reported that high electric fields could enhance the possibility of protein unfolding due to the heterogeneous nature of charge distribution within proteins [29].

In our study, the effects of pH on the propensity of beta-strand formation that might lead to amyloidosis were addressed in atomistic details as the low pH condition was previously reported to facilitate amyloidosis. In addition, the altered electrostatic properties by the protonation of some amino acids should affect protein denaturation and beta-strand formation. Firstly, a series of MD simulations were performed under a high temperature at both pH conditions to accelerate unfolding processes, followed by simulations under an optimum temperature for amyloidosis reported in previous studies. Then, the conformational analysis was performed to characterize the simulated protein structures and provide the detailed mechanisms of beta-sheet formation under low pH and molecular insight of accelerating amyloidosis *in vitro*.

## 2. Methodology

GROMACS 2019.6 simulation package was used to carry out the Molecular Dynamics simulations. The initial atomistic structure of hen-egg white lysozyme was obtained from Protein Data Bank (PDB ID: 1AKI). In order to emulate pH conditions, protein coordinates were input to the PropKa software [35, 36] to estimate pKa values for aspartic acid, glutamic acid, and histidine residues. If the specified pH was lower than the pKa of a residue, the protonation state was assigned to that residue. At the pH 2 condition, all histidines, glutamic acids, and aspartic acids were fully protonated. Meanwhile, at pH 7, all the aforementioned amino acids were deprotonated. All protein structures were parameterized by the GROMOS54a7 forcefield [37–39], suitable for the prediction of protein unfolding under high temperatures [37]. Then, the lysozyme structures at both pH conditions were solvated by using the SPC water model [38, 39] within simulation boxes of size 6.9 x 6.9 x 6.9 nm$^3$, which was large enough to cover a whole protein molecule with 1 nm buffer distance. As HEWL in both protonated and deprotonated forms were positively charged, all systems were neutralized by adding Cl- counter-ions. For each simulation, the energy minimization was performed with the steepest descent algorithm for the maximum number of steps of 50000. The short-range electrostatic cutoff distance and the short-range Van der Waals cutoff distance were 1.0 nm. PME treatment was used to calculate long-range electrostatic interactions. Then, an NPT equilibration stage was performed at T = 300 K and P = 1 bar for 200 ns, which was long enough to accommodate the conformation changes from protonation states introduced at pH 2. After that, three replicas of MD simulations at 450 K were performed for 100 ns to accelerate unfolding processes of both protonated (pH 2) and deprotonated (pH 7) structures. Then, all six simulations were continued at 333 K for 200 ns to emulate the *in vitro* condition that beta-strand formation was observed.

All MD trajectory replicas at both pH conditions were then analysed. Root mean square deviation (RMSD) was calculated to quantify the level of global conformational changes compared with the reference PDB structure. The time-dependent information of the secondary structure content of lysozymes at both pH 2 and pH 7 were analyzed by the DSSP algorithm [40], which identified the type of secondary structure for all regions within the protein. During the unfolding process at 450 K and the refolding processes at 333 K, the states of backbone torsions and alpha-beta transitions were monitored by Ramachandran plots for all simulation replicas at both pH conditions. The radius of gyration (Rg) was calculated as a function of time for different amino acid groups of the HEWL to monitor the distribution of charged amino acids and hydrophobic amino acids.

## 3. Results and discussions

A series of atomistic molecular dynamics (MD) simulations on HEWL under two different pH conditions and conformational analysis of all simulation replicas were performed to understand the effect of low pH on the unfolding of HEWL and the propensity of beta-strand formation. For each replica, HEWL was explicitly simulated at 450 K for 100 ns to accelerate the unfolding process within a feasible timescale for MD simulations, followed by a 200-ns run at 333 K temperature to emulate the condition where amyloidosis was observed *in vitro*. In Fig 1, the root mean square deviation (RMSD) of HEWL was calculated from all MD trajectories to monitor the global conformational changes of HEWL, along with the snapshots at the start, the middle (50 ns), and the end (100 ns) of the 450-K unfolding simulation. For the pH 2 condition at 450 K (Fig 1A), the RMSD values of lysozyme in all repeats increased rapidly within the first 30 ns. The replica R0 displayed the most abrupt change as the RMSD exceeded 1.2 nm after 10 ns, while the RMSD of replicas R1 and R2 increased beyond 1.2 nm after 30 ns and 20

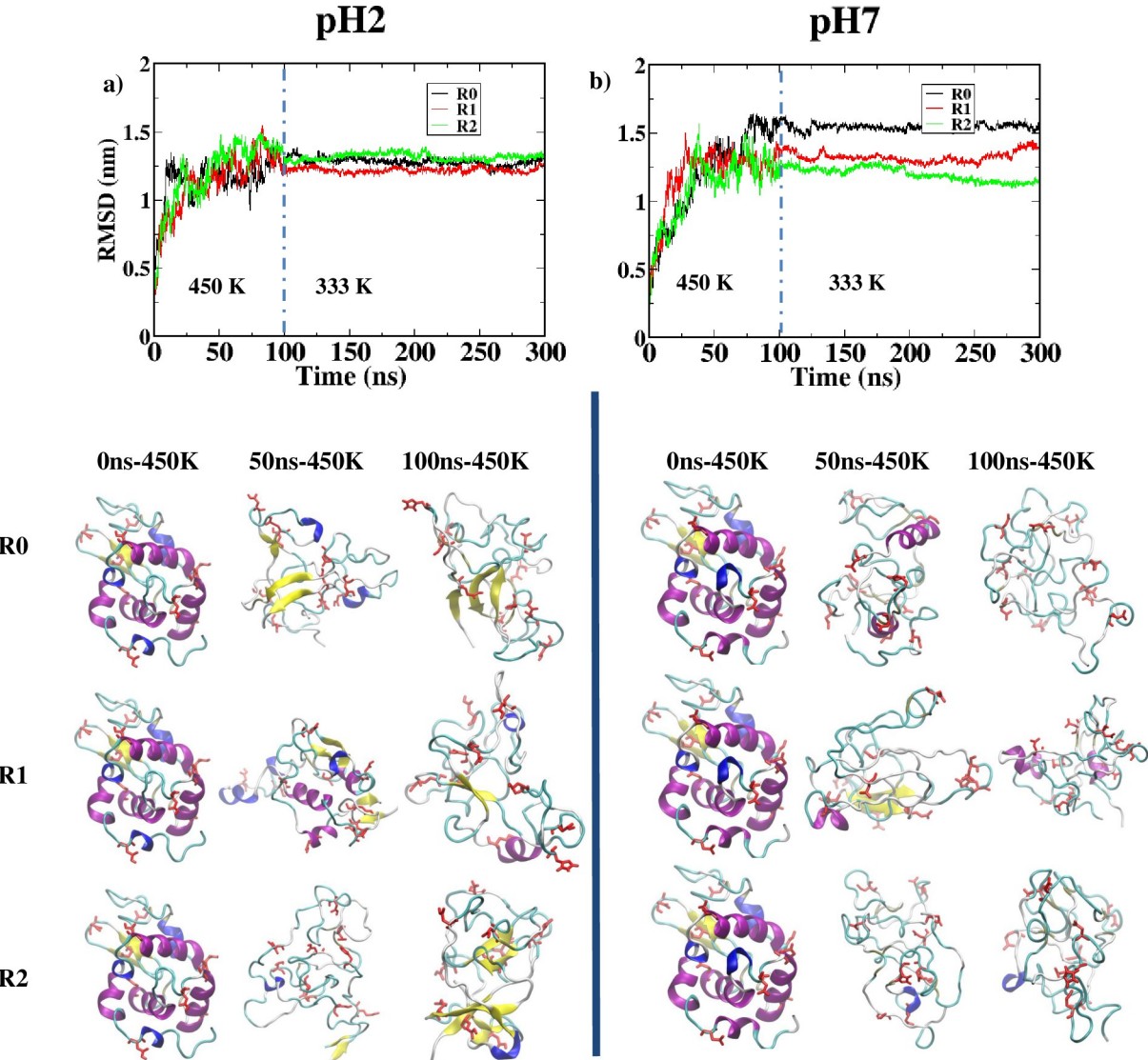

**Fig 1.** RMSD calculated along all three replicas of MD simulations at 450 K and 333 K–R0 (black), R1 (red), and R2 (green)—at (a) pH 2 and (b) pH 7. Vertical dashed lines represent the time t = 100 ns where the temperature was switched from 450 K to 333 K. Conformational snapshots after 0 ns, 50 ns, and 100 ns of 450 K simulations were also presented for each replica, in which alpha helices were represented in purple and beta strands were represented in yellow.

ns, respectively. The conformational snapshots of HEWL replicas showed that most of the alpha-helical structures disappeared in R0 and R2 after 50 ns. Only one short alpha-helix was found for the R1 replica at 450 K and pH 2 after 100 ns. Interestingly, additional beta-strands were formed in all three replicas, especially in R0 and R2. The averaged RMSD from all replicas at 450 K and pH 2 after 50 ns was 1.35 ± 0.15 nm. Less fluctuation was then observed for the RMSD values when the systems at pH 2 were cooled down to 333 K, and the averaged RMSD was found at 1.30 ± 0.10 nm. For simulations at pH 7 and 450 K (Fig 1B), RMSD of the replicas R0, R1, and R2 exceeded 1.2 nm after 35 ns, 20 ns, and 30 ns, respectively. Similar to the pH 2 simulations, most of the alpha-helices of HEWL became lost during 450 K simulations. However, more helical structures were still left at pH 7 as two short helices were observed for the R1 replica, while one helical structure was observed for the R0 and R2 replicas. No beta-strand

was observed after 100 ns for all pH 7 simulations. Averaged RMSD at pH 7 and 450 K after 50 ns was found around $1.30 \pm 0.15$ nm and became less fluctuating when the temperature was decreased to 333 K. Transition of alpha helices to either beta strands or random coils could also be visualized through the Ramachandran plots (see S1 and S2 Figs in S1 File).

RMSD results from 450 K simulations at pH 2 and 7 suggested that HEWL unfolded faster at pH 2, incorporating additional beta-strand formation. The DSSP (Define Secondary Structure of Protein) algorithm could quantitatively measure the secondary structure content for all simulations by identifying the types of structures for all the amino acid residues from patterns of hydrogen bonding network. Time evolution of the secondary structure content within the simulated HEWL at pH 2 (Fig 2) showed that most of the alpha-helical structures were lost during the 100-ns simulations at 450 K. In the case of the R0 replica at pH 2, three of the four alpha-helices were denatured after only 10 ns, corresponding to the most rapidly increasing RMSD. The R2 replica displayed similar behavior at the early 450 K stage but with a slightly longer timescale than the R0 replica. For the R1 replica, an alpha-helix of the sequence TASVNCAKKIVS at residues 89–100 could withstand the 450 K temperature up to 90 ns. The same helical region was also the last alpha-helix to denature for the R0 and R2 replicas. Beta strand formation and deformation occurred at different times and locations under 450 K for all replicas at pH 2. After the temperature was switched from 450 K to 333 K at 100 ns, the regions identified as beta-strands or at the beginning of 333 K simulations were extended, corresponding to the beta-strands observed in the final snapshots of all three replicas for pH 2 simulations. Only short beta-strands were found for the replica R1, while a long beta-strand of sequence VCAAKFE at residues 29–35 was found for the replica R2. Meanwhile, the more extended SLGNWVCAAKFES beta-strand at residues 24–36 was found for the replica R0 about the same region, along with the YGILQINSRWW beta-strand at residues 53–63. The time evolution of the secondary structures of simulated HEWL at pH 7 (Fig 3) showed significantly less amount of beta-strand formed after the temperature switching at 100 ns. On the other hand, alpha-helices were formed at the 333 K simulation stages of the R1 and R2 replicas around the region containing residues 89–100, where the alpha helix of sequence TASVNCAKKIVS displayed high thermostability at both pH conditions. Another alpha-helix near the N-terminus was formed during the 200-ns simulation at 333 K and pH 7 for the replica R1.

Tables 1 and 2 summarized the amount of alpha-helix and beta-strand secondary structures for all simulation replicas at both pH conditions. The DSSP analysis on the HEWL structures after a 200-ns equilibration at 300 K showed that the alpha-helix content within the native HEWL was 32.5% of the whole structure at pH 2, and 34.1% at pH 7. Meanwhile, the beta-sheet or beta-strand content within the HEWL was 6.2% of the whole structure at both pH conditions. After all the 300-ns simulations finished, the final percentage of alpha-helix content was found between 0.0% - 4.4% at pH2 and 0.1% - 13.0% at pH 7, suggesting that the full protonation on HEWL at pH 2 corresponded to more alpha helix loss. However, the percentage of beta-sheet content was found between 9.1% - 19.1% at pH 2 and 0.6% - 3.8% at pH 7, confirming that alpha-beta transition was more likely to occur at pH 2.

To further consider the effects of pH on the beta-strand formation of the simulated HEWL, the radius of gyration (Rg) was determined from different groups of amino acids. The distribution of positively-charged (Arg, Lys, and His) and negatively-charged (Glu and Asp) amino acids was affected by the protonation of all eight Glu and Asp residues at pH 2, which neutralized all the negatively-charged sidechains. Meanwhile, the Rg was calculated from all hydrophobic amino acids to monitor the propensity to form clusters. Fig 4 displayed the radius of gyration calculated along with all the 333 K simulations at both pH conditions for groups of hydrophobic amino acids, positively-charged amino acids (ARG-LYS-HIS), and negatively-charged amino acids (GLU-ASP). For all simulations at pH 2 (Fig 4A and 4C), Rg of the

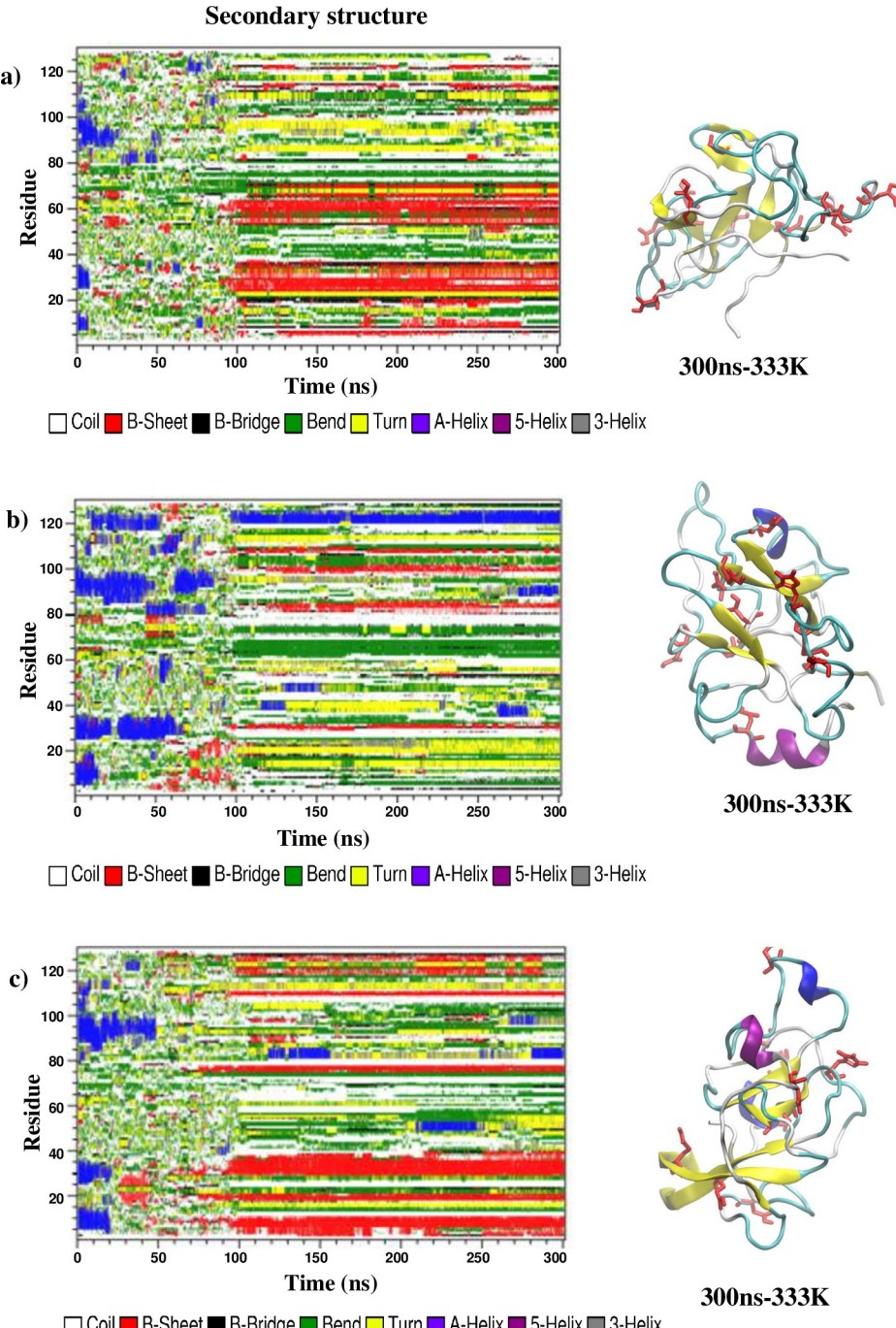

**Fig 2.** Time evolution of the secondary structures contents determined by the DSSP algorithm along the 450-K and 333-K simulation replicas (a) R0, (b) R1, and (c) R2 at pH 2. Snapshots of the final structure after the 333 K simulation was also given for each replica.

positively-charged ARG-LYS-HIS group was found above the Rg of the whole HEWL, while Rg of the groups of hydrophobic residue was found below the Rg of the whole HEWL. Rg of the protonated GLU-ASP group was located above the Rg of the whole HEWL for the R0 replica and was found below the Rg of the whole HEWL for the R1 and R2 replicas. At pH 7

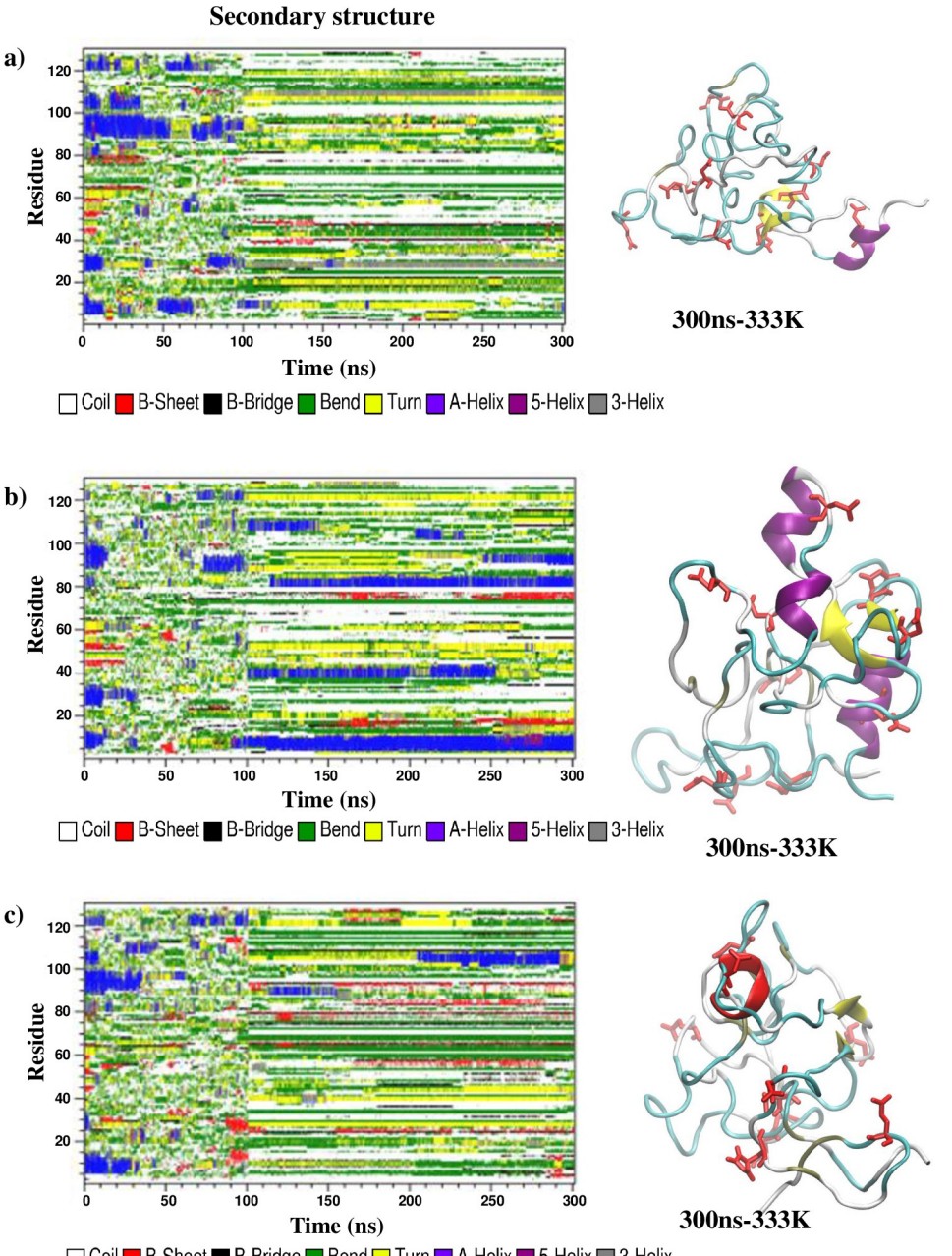

**Fig 3.** Time evolution of the secondary structures contents determined by the DSSP algorithm along the 450-K and 333-K simulation replicas (a) R0, (b) R1, and (c) R2 at pH 7. Snapshots of the final structure after the 333 K simulation was also given for each replica.

(Fig 4D and 4F), Rg of the positively-charged ARG-LYS-HIS group was found slightly above the Rg of the whole HEWL, while the only slight difference in Rg was observed for the negatively-charged GLU-ASP group, hydrophobic group, and the whole HEWL.

The significant difference between the Rg of the group of hydrophobic residues and the group of positively-charged residues (ARG-LYS-HIS) at pH 2 signified that hydrophobic residues tended to form a hydrophobic cluster within HEWL cores. In contrast, the positively charged residues tended to be more exposed to the solution. Final snapshots highlighting C-

**Table 1. The percentage of alpha-helix and beta-strand secondary structures for all simulation replicas at pH 2.**

| pH2 replicas | Structure | 0 ns (%) | 50 ns (%) | 100 ns (%) | 300 ns (%) |
|---|---|---|---|---|---|
| R0 | α | 32.5 | 4.1 | 0.1 | 0 |
| | β | 6.2 | 5.5 | 11.9 | 16 |
| R1 | α | 32.5 | 20.9 | 3.3 | 3.5 |
| | β | 6.2 | 4.8 | 7.1 | 9.1 |
| R2 | α | 32.5 | 0.8 | 0.2 | 4.4 |
| | β | 6.2 | 3.6 | 16.1 | 19.1 |

alpha atoms of hydrophobic residues in Fig 4A and 4C showed that the clusters of hydrophobic residues were presented along with the formed beta strands at pH 2. At pH 7, as Glu and Asp residues were deprotonated and the total charge of HEWL was reduced from +16e to +8e, the difference between the Rg of hydrophobic and charged groups became less significant. As a result, hydrophobic became more exposed to the solution and less likely to form a cluster at pH 7. The relationship between hydrophobicity and beta-strand formation could be discussed regarding the beta-strand sequences observed from simulations. Table 3 summarized the regions where beta-strands were found in final structures at pH 2, along with their amino acid sequences. Beta strands from final structures of all simulations at pH 2 involved 57 amino acids, of which 28 of them (49.12%) contained hydrophobic sidechains. The hydrophobic content within the beta-strands was clearly higher than the 35.66% of hydrophobic content within the whole HEWL. The YGILQINSRWW beta-strand found in the R0 replica at pH 2 contained 54.55% of hydrophobic content, even higher than the average hydrophobic content for beta-strands. This YGILQINSRWW sequence at residues 53–63 in one of our simulations covered the K-peptide region (residues 54–62) previously proposed as the core for amyloidosis [41–44]. An experimental study by Tokunaga et al. later showed that K-peptides were most likely to form fibrils under acidic conditions. Adding the STDY sequence from the residues 50–53 of HEWL further enhanced the amyloidosis under low pH [45].

## 4. Conclusions

In this study, the beta-strand formation mechanism at the early stage of amyloid fibrilization has been proposed. Lysozyme is one of the amyloid proteins that can misfold into fibrils and cause some diseases. However, the controlled synthetic amyloid fibrils can become useful biomaterials for bioengineering and biosensing applications. Our molecular dynamics simulations showed that beta-strands were more likely to form when HEWL was unfolded at pH 2. Mechanisms of beta-strand formation were explained by the radial distribution of charged and the hydrophobic amino acids. The beta-strand forming sequences from the HEWL simulations contained a significantly higher hydrophobic content than the whole HEWL. This result was consistent with the relationship between hydrophobic clustering and amyloidosis

**Table 2. The percentage of alpha-helix and beta-strand secondary structures for all simulation replicas at pH 7.**

| pH7 replicas | Structure | 0 ns (%) | 50 ns (%) | 100 ns (%) | 300 ns (%) |
|---|---|---|---|---|---|
| R0 | α | 34.1 | 11.9 | 5.9 | 0.9 |
| | β | 6.2 | 0.5 | 1.4 | 0.6 |
| R1 | α | 34.1 | 2.6 | 8.3 | 13.2 |
| | β | 6.2 | 6.5 | 0.4 | 3.8 |
| R2 | α | 34.1 | 2.2 | 4.3 | 0.1 |
| | β | 6.2 | 4.5 | 8.7 | 3.1 |

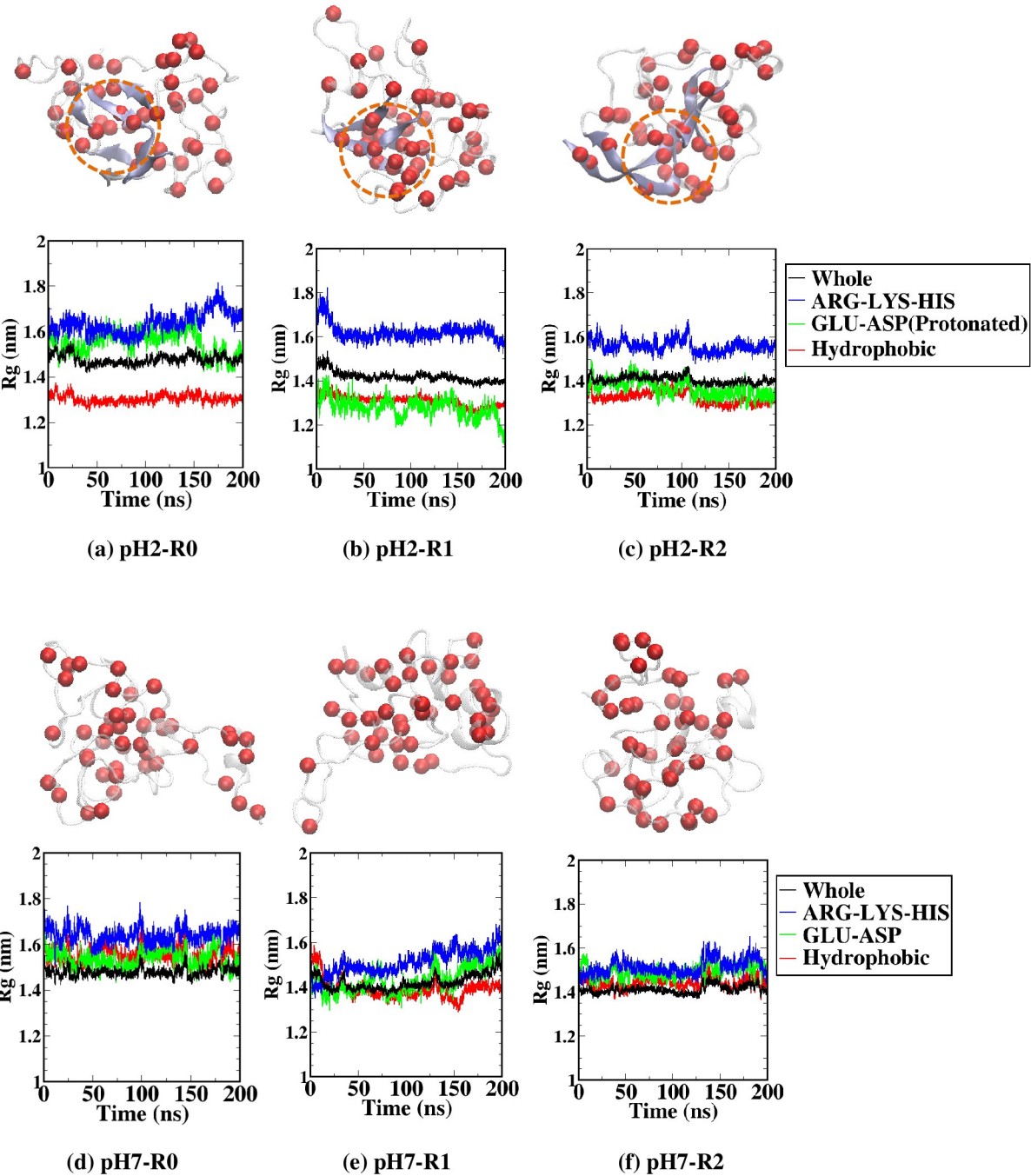

**Fig 4.** Radius of gyration (Rg) calculated for the groups of whole HEWL structure (black), positively-charged residues (ARG-LYS-HIS; blue), negatively-charged residues (GLU-ASP; green), and hydrophobic residues (red) along all three 333-K simulation replicas of HEWL at (a-c) pH 2 and (d-f) pH 7. The negatively-charged residues were protonated at pH 2. Final snapshots highlighting C-alpha atoms of hydrophobic residues (red sphere) were also shown. Dash circles indicated the cluster of hydrophobic residues around the beta-strand regions at pH 2.

previously reported by Mishima et al. [46]. Furthermore, at pH 2, a significant positive charge added by the protonation of all glutamic acid and aspartic acid residues corresponded with the separation of positively-charged amino acids from the group of hydrophobic residues, which

**Table 3. Residue numbers and amino acid sequences of beta-strand found in the final structure of HEWL from all simulation replicas at pH 2.**

| pH 2 Replica | Residue number of Beta |
|---|---|
| R0 | 24–36 (SLGNWVCAAKFES), 53–63 (YGILQINSRWW), 65–66 (ND), 69–70 (TP), 103–104 (NG), 112–113 (RN) |
| R1 | 29–31 (VCA), 83–85 (LLS), 99–101 (VSD), 107–109 (AWV) |
| R2 | 2–3 (VF), 6–10 (ELAA), 18–20 (DNY), 29–35 (VCAAKFE), 38–39 (FN), 75–77 (LCN), 109–110 (VA) |

promoted both hydrophobic clustering and beta-strand formation. However, further validation of this mechanistic scheme is still needed for the beta-strand formation of other amyloid proteins towards the control of fibril production and the prediction of nucleation sites.

## Supporting information

**S1 File. Supplementary information file contains Ramachandran plots at pH 2 and pH 7.** (DOCX)

## Acknowledgments

This research project is supported by Thailand Science Research and Innovation (TSRI) Basic Research Fund: Fiscal year 2022 (FF65). HFZ was funded by the Petchra Pra Jom Klao Doctoral Scholarship, KMUTT.

## Author Contributions

**Conceptualization:** Thana Sutthibutpong.

**Data curation:** Husnul Fuad Zein, Ibrar Alam, Piyapong Asanithi.

**Formal analysis:** Husnul Fuad Zein.

**Funding acquisition:** Husnul Fuad Zein, Thana Sutthibutpong.

**Investigation:** Thana Sutthibutpong.

**Software:** Thana Sutthibutpong.

**Supervision:** Piyapong Asanithi, Thana Sutthibutpong.

**Visualization:** Husnul Fuad Zein.

**Writing – original draft:** Husnul Fuad Zein.

**Writing – review & editing:** Ibrar Alam, Piyapong Asanithi, Thana Sutthibutpong.

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
