## [Decision Letter · Decision Letter 0]

14 Jun 2021

PONE-D-21-08882

Molecular Dynamics Study on the Effects of Charged Amino Acid Distribution Under low pH Condition to the Unfolding of Hen Egg White Lysozyme and Formation of Beta Strands.

PLOS ONE

Dear Dr. Sutthibutpong,

Thank you for submitting your manuscript to PLOS ONE. After careful consideration, we feel that it has merit but does not fully meet PLOS ONE’s publication criteria as it currently stands. Therefore, we invite you to submit a revised version of the manuscript that addresses the points raised during the review process.

We look forward to receiving your revised manuscript.

Kind regards,

Human Rezaei

Academic Editor

PLOS ONE

Journal Requirements:

Reviewers' comments:

Reviewer's Responses to Questions

**Comments to the Author**

1. Is the manuscript technically sound, and do the data support the conclusions?

Reviewer #1: Partly

2. Has the statistical analysis been performed appropriately and rigorously? 

Reviewer #1: No

3. Have the authors made all data underlying the findings in their manuscript fully available?

Reviewer #1: Yes

4. Is the manuscript presented in an intelligible fashion and written in standard English?

Reviewer #1: Yes

5. Review Comments to the Author

Reviewer #1: In this paper, Zein et al provide molecular dynamics simulations of Hen egg white lysozyme partial denaturation and refolding at high temperature. The protocol they present is a 100-ns simulation at temperature 450 K followed by 200 ns at 333 K. The authors use two protonation states, one where all titrable side chains (including glutamate, aspartate, histidine) are protonated ('pH 2'), and one where glutamates, aspartates and histidines are deprotonated ('pH 7'). They make three replicas of each state and analyze several parameters along the simulations : RMSD, secondary structures and various radii of gyration. The actual content of the paper is small and quite incremental with respect to existing data, both experimental and computational, on lysozyme unfolding and refolding and on amyloidosis. Still the authors make one convincing and interesting observation: That under their protocol protonated lysozyme consistently transitions from alpha-helix- to beta-sheet-containing structures, while there is no beta-sheet formation in the deprotonated form. A shortened paper focusing on this observation would be more suitable for publication.

Pages are not numbered so the following refers to the pages of the consolidated pdf, with the introduction starting on page 9.

1) p. 10 'The accuracy of atomistic molecular dynamics (MD) simulations to predict the

molecular behavior of proteins under extreme conditions [24,25] has been improved with the

continuing development of molecular mechanics forcefield parameters [26,27].'

I agree with this, but the authors should at least indicate in the methods which forcefield they used, and if there was a reason (suitability for high temperature ?) for the choice.

Similarly p.11 'the SPC216 water model': Was there a reason for this water model ?

2) 'Table 1: All simulations in this study' is not useful. As stated there are just two nearly identical systems simulated in the same way with three replicas each. Please remove.

3) p. 11 'At the pH 2 condition, all histidines, glutamic

acids, and aspartic acids were fully protonated. Meanwhile, at pH 7, all the aforementioned amino acids were deprotonated.'

Was the C-terminus considered ?

4) p. 14

pH 2 'At this stage, the slight drop of RMSD values denoted the

refolding of proteins.'

pH7 : 'No significant drop of RMSD was seen when switching the temperature to 333 K, and hence

no protein refolding.'

a) This is not obvious from the RMSD and analyses provided. The parts about protein refolding are not substantiated and should be removed.

b) Generally speaking the results are full of discussion elements and speculations such as these. The manuscript thus can and should be drastically shortened (see below points 8 and 9).

5) p. 15 'The transitions from alpha-helixes, which were the major part of native lysozymes, into

beta-strands and random coils were quantified through the Ramachandran plot'

There is no quantification given from figures 2 and 3 (Ramachandran plots). Quantifications are given from figures 4 and 5 (DSSP plots) only. All elements given in the Ramachandran part are qualitative, such as :

'The alpha-beta transitions occurred less frequently for the non-protonated proteins at

pH7 (Figure 3), in which a large number of amino acid'

This part should be shortened, removing particularly interpretations such as 'signified the incomplete refolding process as observed from the conformational snapshots'

6) DSSP plots

p. 16 'the percentage of beta-sheet content was

found between 9.1% - 19.1% at pH 2 and 0.6% - 3.8% at pH 7,'

This is the interesting and substantial result the authors bring and should be the focus of the results section. As it is, it is clear from the DSSP plots that the same residues are not found in the same conformation in different replicas. This should at least be stated in the text. Preferably residues that tend to be incorporated in beta-strands should de identified, in relation to the 'radii of gyration' plots (see below).

7) Radii of gyration

That part is the least clear and convincing of the paper. A major problem is that it is not clear which selections statements such as 'negatively-charged

amino acids affected by the protonations.' 'positively charged, negatively-charged, and hydrophobic compositions' refer to.

a) I take it from Figure 6 and 7 labels ('Negative-protonated' instead of 'Negative') that the same selections (namely asp+glu) are considered at pH 2 and pH 7 ? Please then refer explicitly to these as 'asp+glu', both in the text and the figures.

b) What about 'positive' residues ? Are histidine in the 'positive' selection at pH 2 and pH 7 ? Only at pH 2 ? 'Rg of the positively charged group was the highest Rg due to a large number of positive charge residues.' suggests that it is the latter. But then, no meaningful comparison can be made between 'positive' radii of gyration at pH 7 vs pH 2 (different selections). At any rate, please also make explicit descriptions of the 'positive' selection (whether arg+lys or arg+lys+his).

c) p. 17 'However, at pH 7

(Figure 6d, 6e and 6f), Rg values of different amino acid groups tended to converge as the

simulation progressed.'

This is simply false. Only for R1 is it verified, for R0 and R2 'positive' radii stay above others as for pH 2.

d) p. 17

'the formation of beta-strands

or beta-sheets from backbone parts of hydrophobic amino acids was facilitated by the increased

compactness (low Rg) of the hydrophobic clusters'

No results are given as to the composition of beta strands (see point 6 above) and whether they tend to be made of hydrophobic residues. Figure 8 is illustrative at best. The second part of the sentence is interpretation.

8) The whole section '4. Discussions' is an extended repetition of results. For instance

p. 20 'positively charged residues formed a hydrophilic shell with larger Rg than the averaged Rg' (and as stated this particular point is moot anyway since it is also the case at pH 7 for 2 replicas out of 3). Furthermore, the 'results' section itself is riddled with interpretations. The easiest way to amend this is to remove section '4. Discussions' completely, and change the section '3. Results' into '3. Results and discussion'

9) In this new 'Results and discussion', please avoid speculations and interpretations that are not substantiated by the results themselves. For instance

a) p. 20 'The higher amount of Coulombic repulsion at

lower pH had driven most of the positively charged sidechain further from the backbone,

leaving the backbone to stay at the middle between hydrophobic and hydrophilic shells. The

beta-strands were finally formed by nucleation of the ordered backbone part.'

No analyses are provided to support this, not even the justification of a visual inspection of the backbone.

b) p. 18

'to refold into betastrands

at pH 2, while misfolded into alpha-helices and random coils at pH 7'

Why speak of 'refolding' into beta-strands and 'misfolding' into alpha-helices for an alpha-helical protein ?

10)Minor: There are many typos throughout the manuscript such as'to observed the effects', 'analyzedanalyzed by the DSSP algorithm'. Please proofread carefully.

6. PLOS authors have the option to publish the peer review history of their article (what does this mean?). If published, this will include your full peer review and any attached files.

Reviewer #1: No

---

## [Author Response · Author response to Decision Letter 0]

30 Jul 2021

We thank the Editor-in-chief for giving us an opportunity to revise the manuscript and the reviewer for his/her comments and suggestions. The paper is revised according to comments and suggestions of the reviewer and we have highlighted the changes made in the revised manuscript with the red color.

Reviewer 1’s: In this paper, Zein et al provide molecular dynamics simulations of Hen egg white lysozyme partial denaturation and refolding at high temperature. The protocol they present is a 100-ns simulation at temperature 450 K followed by 200 ns at 333 K. The authors use two protonation states, one where all titrable side chains (including glutamate, aspartate, histidine) are protonated ('pH 2'), and one where glutamates, aspartates and histidines are deprotonated ('pH 7'). They make three replicas of each state and analyze several parameters along the simulations : RMSD, secondary structures and various radii of gyration. The actual content of the paper is small and quite incremental with respect to existing data, both experimental and computational, on lysozyme unfolding and refolding and on amyloidosis. Still the authors make one convincing and interesting observation: That under their protocol protonated lysozyme consistently transitions from alpha-helix- to beta-sheet-containing structures, while there is no beta-sheet formation in the deprotonated form. 

A shortened paper focusing on this observation would be more suitable for publication. Pages are not numbered so the following refers to the pages of the consolidated pdf, with the introduction starting on page 9.

Comment 1: 

p. 10 (p.5 in the new version) 'The accuracy of atomistic molecular dynamics (MD) simulations to predict the molecular behavior of proteins under extreme conditions [24,25] has been improved with the continuing development of molecular mechanics forcefield parameters [26,27].' I agree with this, but the authors should at least indicate in the methods which forcefield they used, and if there was a reason (suitability for high temperature ?) for the choice. 

Similarly p.11 'the SPC216 water model': Was there a reason for this water model ?

Response: Our choice of forcefield is GROMOS54a7, which has been added into the methodology section (p.5 in the new version). The reason behind choosing this forcefield comes from a benchmarking study (also cited in the paper) of Kamenik et al. 2020, where proteins simulated by GROMOS54a7 forcefield showed a good agreement with experimental state population and unfolding time. Then, the SPC water model was used along with the GROMOS forcefield as the GROMOS forcefield itself was parameterized to reproduce the free enthalpy of hydration in SPC water (‘SPC216’ was the name of the file so we changed it back to the actual name of the model ‘SPC’). Even if SPC is known to incorrectly reproduce the nature of lipid bilayer, there should be no problem as the lipid bilayers do not exist in our models.

Comment 2:

'Table 1: All simulations in this study' is not useful. As stated there are just two nearly identical systems simulated in the same way with three replicas each. Please remove.

Response: The table has been removed.

Comment 3:

p. 11 'At the pH 2 condition, all histidines, glutamic acids, and aspartic acids were fully protonated. Meanwhile, at pH 7, all the aforementioned amino acids were deprotonated.' Was the C-terminus considered ?

Response: We missed out considering the terminus but we believe that it will not significantly affect the protein that already contains eight protonable sites.

Comment 4:

p. 14 pH 2 'At this stage, the slight drop of RMSD values denoted the refolding of proteins.' pH7 : 'No significant drop of RMSD was seen when switching the temperature to 333 K, and hence no protein refolding.' a) This is not obvious from the RMSD and analyses provided. The parts about protein refolding are not substantiated and should be removed. b) Generally speaking the results are full of discussion elements and speculations such as these. The manuscript thus can and should be drastically shortened (see below points 8 and 9).

Response: a) We removed all explanations about protein refolding from the manuscript, but chose to present the events as the formation of beta-strands. RMSD results were discussed along with the snapshot of the HEWL at the start, 50 ns, and 100 ns of 450-K simulations to monitor the conformational change of the proteins at different pH (see Figure 1 and p.7). b) In this version, we already tried to avoid speculative assumptions, e.g. protein refolding after the temperature switching and Coulombic force that drive positively charged residues away. These changes are described more in the comments below.

Comment 5:

p. 15 'The transitions from alpha-helices, which were the major part of native lysozymes, into beta-strands and random coils were quantified through the Ramachandran plot' There is no quantification given from figures 2 and 3 (Ramachandran plots). Quantifications are given from figures 4 and 5 (DSSP plots) only. All elements given in the Ramachandran part are qualitative, such as : 'The alpha-beta transitions occurred less frequently for the non-protonated proteins at pH7 (Figure 3), in which a large number of amino acid' This part should be shortened, removing particularly interpretations such as 'signified the incomplete refolding process as observed from the conformational snapshots'

Response: We discussed more about quantitative measurement on the secondary structure from DSSP results (Table 1 and Table 2) including the percentage, and the amino acid residues that formed beta strand in the final structure (see Table 3). And we removed the discussion about the Ramachandran plot, as it was already covered by the DSSP result - which should shorten the manuscript.

Comment 6:

DSSP plots p. 16 'the percentage of beta-sheet content was found between 9.1% - 19.1% at pH 2 and 0.6% - 3.8% at pH 7,' This is the interesting and substantial result the authors bring and should be the focus of the results section. As it is, it is clear from the DSSP plots that the same residues are not found in the same conformation in different replicas. This should at least be stated in the text. Preferably residues that tend to be incorporated in beta-strands should be identified, in relation to the 'radii of gyration' plots (see below).

Response: We have provided the tables that summarized the percentage of all replicas and the amino acid residues that tend to form beta-strands. We also discussed what happened for each replica at both pH (see p.8 in the new version), the sequences of all beta-strands formed and their hydrophobicity (see p.10 in the new version). As the beta-strand forming sequences contained more hydrophobic residues than other regions. Hydrophobic residues tends to form clusters, resulting in the enhanced compactness but lower radius of gyration (Rg) 

Comment 7:

Radii of gyration That part is the least clear and convincing of the paper. A major problem is that it is not clear which selections statements such as 'negatively-charged amino acids affected by the protonations.' 'positively charged, negatively-charged, and hydrophobic compositions' refer to.

a) I take it from Figure 6 and 7 labels ('Negative-protonated' instead of 'Negative') that the same selections (namely asp+glu) are considered at pH 2 and pH 7 ? Please then refer explicitly to these as 'asp+glu', both in the text and the figures.

Response: Negatively charged amino acids refer to GLU and ASP amino acids at both pH. At pH2 GLU and ASP amino acid were affected by the protonation of all eight Glu and Asp residues. 

b) What about 'positive' residues ? Are histidine in the 'positive' selection at pH 2 and pH 7 ? Only at pH 2 ? 'Rg of the positively charged group was the highest Rg due to a large number of positive charge residues.' suggests that it is the latter. But then, no meaningful comparison can be made between 'positive' radii of gyration at pH 7 vs pH 2 (different selections). At any rate, please also make explicit descriptions of the 'positive' selection (whether arg+lys or arg+lys+his).

Response: Positive residues refer to ARG, LYS, and HIS and were mentioned as ARG-LYS-HIS in the Figure 4 and the whole manuscript.

c) p. 17 'However, at pH 7 (Figure 6d, 6e and 6f), Rg values of different amino acid groups tended to converge as the simulation progressed.' This is simply false. Only for R1 is it verified, for R0 and R2 'positive' radii stay above others as for pH 2.

Response: We have now removed this discussions. In Figure 4 of the revised version, we discussed only the Rg from 333 K simulations. At pH 7, Rg of every groups was not significantly different. Meanwhile, the significant difference in Rg between the positively charged group and the hydrophobic group suggested that the positively charged group was separated from the hydrophobic core. This was discussed in p.10 of the revised version

d) p. 17 'the formation of beta-strands or beta-sheets from backbone parts of hydrophobic amino acids was facilitated by the increased compactness (low Rg) of the hydrophobic clusters' No results are given as to the composition of beta strands (see point 6 above) and whether they tend to be made of hydrophobic residues. Figure 8 is illustrative at best. The second part of the sentence is interpretation.

Response: In the revised version, %content of hydrophobic residues within the beta-strands was analysed in comparison with the whole HEWL structure (p.10 of the new version). Beta-strands contained more hydrophobic content than the whole HEWL structure. 

Comment 8:

The whole section '4. Discussions' is an extended repetition of results. For instance p. 20 'positively charged residues formed a hydrophilic shell with larger Rg than the averaged Rg' (and as stated this particular point is moot anyway since it is also the case at pH 7 for 2 replicas out of 3). Furthermore, the 'results' section itself is riddled with interpretations. The easiest way to amend this is to remove section '4. Discussions' completely, and change the section '3. Results' into '3. Results and discussion'

Response: We have now merged section ‘3. Result’ and ‘4. Discussion’ into section ‘3. Result and Discussion’. 

Comment 9:

In this new 'Results and discussion', please avoid speculations and interpretations that are not substantiated by the results themselves. For instance

a) p. 20 'The higher amount of Coulombic repulsion at lower pH had driven most of the positively charged sidechain further from the backbone, leaving the backbone to stay at the middle between hydrophobic and hydrophilic shells. The beta-strands were finally formed by nucleation of the ordered backbone part.' No analyses are provided to support this, not even the justification of a visual inspection of the backbone.

Response: We left out the discussion on the Coulombic interactions but only mentioning the separation between positively-charged amino acids from the hydrophobic core (judging from significant difference in Rg at pH 2; p.10 of the revised version). We also tried to visualize the tendency for hydrophobic clustering in Figure 4. 

b) p. 18 'to refold into betastrands at pH 2, while misfolded into alpha-helices and random coils at pH 7' Why speak of 'refolding' into beta-strands and 'misfolding' into alpha-helices for an alpha-helical protein ?

Response: We already left out this statement, only reporting time-dependent secondary structure content from DSSP.

Comment 10:

Minor: There are many typos throughout the manuscript such as 'to observed the effects', 'analyzedanalyzed by the DSSP algorithm'. Please proofread carefully.

Response: The draft has been thoroughly proofread. Thank you very much.

---

## [Decision Letter · Decision Letter 1]

24 Feb 2022

Molecular Dynamics Study on the Effects of Charged Amino Acid Distribution Under low pH Condition to the Unfolding of Hen Egg White Lysozyme and Formation of Beta Strands.

PONE-D-21-08882R1

Dear Dr. Sutthibutpong,

We’re pleased to inform you that your manuscript has been judged scientifically suitable for publication and will be formally accepted for publication once it meets all outstanding technical requirements.

Kind regards,

Hannes C Schniepp, Dr. sc. nat.

Academic Editor

PLOS ONE

Additional Editor Comments (optional):

Reviewers' comments:

Reviewer's Responses to Questions

**Comments to the Author**

1. If the authors have adequately addressed your comments raised in a previous round of review and you feel that this manuscript is now acceptable for publication, you may indicate that here to bypass the “Comments to the Author” section, enter your conflict of interest statement in the “Confidential to Editor” section, and submit your "Accept" recommendation.

Reviewer #1: All comments have been addressed

2. Is the manuscript technically sound, and do the data support the conclusions?

Reviewer #1: Yes

3. Has the statistical analysis been performed appropriately and rigorously? 

Reviewer #1: N/A

4. Have the authors made all data underlying the findings in their manuscript fully available?

Reviewer #1: Yes

5. Is the manuscript presented in an intelligible fashion and written in standard English?

Reviewer #1: Yes

6. Review Comments to the Author

Reviewer #1: (No Response)

7. PLOS authors have the option to publish the peer review history of their article (what does this mean?). If published, this will include your full peer review and any attached files.

Reviewer #1: No

---

## [Editor Report · Acceptance letter]

15 Mar 2022

PONE-D-21-08882R1 

Molecular Dynamics Study on the Effects of Charged Amino Acid Distribution Under low pH Condition to the Unfolding of Hen Egg White Lysozyme and Formation of Beta Strands. 

Dear Dr. Sutthibutpong:

I'm pleased to inform you that your manuscript has been deemed suitable for publication in PLOS ONE. Congratulations! Your manuscript is now with our production department. 

Kind regards, 

on behalf of

Dr. Hannes C Schniepp 

Academic Editor

PLOS ONE